cognition/behaviour/psychology

tool use, problem solving, physical cognition, cognitive development, cross-cultural psychology, developmental psychology

**Author for correspondence:**
Karri Neldner
e-mail: karri.neldner@uqconnect.edu.au

# A cross-cultural investigation of young children's spontaneous invention of tool use behaviours

Karri Neldner[1], Eva Reindl[2,3], Claudio Tennie[4], Julie Grant[5], Keyan Tomaselli[5] and Mark Nielsen[1,5]

[1]Early Cognitive Development Centre, School of Psychology, University of Queensland, Queensland, Australia
[2]School of Psychology, University of Birmingham, Edgbaston, Birmingham, West Midlands B15 2TT, UK
[3]School of Psychology and Neuroscience, University of St Andrews, St Mary's Quad, St Andrews KY16 9JP, UK
[4]Department of Early Prehistory and Quaternary Ecology, University of Tübingen, Tubingen, Germany
[5]Department of Communication, University of Johannesburg, Auckland Park, South Africa

KN, 0000-0002-8237-5679; ER, 0000-0003-1594-1367;
CT, 0000-0002-5302-4925; JG, 0000-0003-4464-9429;
KT, 0000-0002-2995-0726; MN, 0000-0002-0402-8372

Through the mechanisms of observation, imitation and teaching, young children readily pick up the tool using behaviours of their culture. However, little is known about the baseline abilities of children's tool use: what they might be capable of inventing on their own in the absence of socially provided information. It has been shown that children can spontaneously invent 11 of 12 candidate tool using behaviours observed within the foraging behaviours of wild non-human apes (Reindl *et al.* 2016 *Proc. R. Soc. B* **283**, 20152402. (doi:10.1098/rspb.2015.2402)). However, no investigations to date have examined how tool use invention in children might vary across cultural contexts. The current study investigated the levels of spontaneous tool use invention in 2- to 5-year-old children from San Bushmen communities in South Africa and children in a large city in Australia on the same 12 candidate problem-solving tasks. Children in both cultural contexts correctly invented all 12 candidate tool using behaviours, suggesting that these behaviours are within the general cognitive and physical capacities of human children and can be produced in the absence of direct social learning mechanisms such as teaching or observation. Children in both cultures were more likely to invent those tool behaviours more frequently observed in great ape populations than those less frequently observed, suggesting there is similarity in the level of difficulty

of invention across these behaviours for all great ape species. However, children in the Australian sample invented tool behaviours and succeeded on the tasks more often than did the Bushmen children, highlighting that aspects of a child's social or cultural environment may influence the rates of their tool use invention on such task sets, even when direct social information is absent.

## 1. Introduction

Tools exist in every human culture, although they vary wildly in form and function. Through the mechanisms of observation, imitation and structured forms of teaching, young children readily pick up how to use the tools of their culture, ensuring that these toolkits are maintained across generations [1,2]. The faithful copying of established techniques within a society, particularly by novices and children, combined with the ability to build upon the inventions of others through furthered (copied) innovation and/or copying error (known as the 'ratchet effect'; [2]) are the fundamental drivers of human cumulative culture [3–5].

But what is the baseline capacity of human tool use? To what extent are individuals capable of using tools in the *absence* of direct social influence? This is a particularly challenging aspect of human development to isolate, as we are entrenched within the cultural landscape of our group in development and surrounded by its social agents from birth [2]. However, if young children unfamiliar with certain tool-based problems can spontaneously produce established tool use behaviours when provided with the necessary materials, and this occurs in the absence of direct social information (such as demonstrations or teaching) and across cultural contexts, this suggests that such behaviours might exist within the 'zone of latent solutions' (ZLS), or within the general physical and cognitive abilities of that species [6–8]. Understanding the extent to which young children may invent simple tool behaviours by individual learning, without the need for social direction, can also help identify when such behaviours may have emerged within our evolutionary history, and which living relatives we might share these capacities with.

To examine the roots of human tool use ability, Reindl *et al*. [9] presented 2- to 3.5-year-old children from a metropolitan UK city and a small German town with a range of novel tasks that could be solved using tools. They modelled these tasks upon some of the cultural foraging behaviours observed across wild chimpanzee and orangutan populations: mimicking the obtaining of a food source such as honey from a beehive or termites from a mound [10,11]. These tasks were deemed novel enough (for humans) that young children were unlikely to have encountered them in their everyday lives,[1] but possessing ecologically relevant demands similar to those experienced by our closest living relatives (and presumably the last shared common ancestor of humans and all great apes).[2] Twelve tasks were developed requiring the use of a tool to open a container or break a barrier to gain access to a sticker. It was found that children were capable of reinventing 11 of the 12 selected tool using behaviours—at least two children invented the candidate behaviour on each of these tasks. It has been proposed that for tool using behaviours, independent re-invention by two individuals is sufficient to conclude that the behaviour did not occur by chance (as the likelihood that all necessary events required for successful tool use to emerge by random is very low; [8,9,12]). Only one task in this battery, nut hammering (arguably the most complex form of tool use), did not reach this criterion for success. Children were also more likely to invent those behaviours observed more frequently in studies of wild ape populations, such as dipping sticks into a cavity to retrieve fluids, than those observed less frequently, such as perforating barriers to gain access to a food source. This overlap suggests that the invention of such tool use behaviours might be within the general cognitive repertoire of most, and

[1]Although a minority of individuals may have been exposed to documentaries showing some of the target behaviours within ape populations, or had other prior knowledge of their existence, we expected this effect to be low overall, and so did not expect possible prior exposure to such material to impact group performances overall.

[2]The circumstances in which naive apes living in populations that already express these behaviours in the wild encounter these problems is often in a similarly socially mediated way, where tools have been discarded by others near the problem. Children as well as apes are therefore presented with the appropriate tool (solution) alongside the problem. However, potentially, the wild context contains more stimuli nearby (e.g. different plants, etc.). It is hard to determine how distracting these items may be. In wild apes, the linkages between tool and problem may be less salient. On the other hand, wild apes may have been exposed to the problem numerous times before their invention occurred, while children here receive a single exposure of a few minutes. Despite these potential differences, the investigation into children's invention of these behaviours is the initial step in linking them to those of great apes in a bid to understand whether shared cognitive capacities allowing (and shaping) such inventions exist. Note that our studies do not intend to recreate the invention behaviours of apes in populations that do not yet express such behaviour.

possibly all, great ape species, lending support to the notion that the capacity for tool use may have evolved within the last shared common ancestor of all great apes, some 14 Ma [13].

Being a successful tool user requires several cognitive and physical capacities [14]. Firstly, an individual must recognize objects for their potential to act as tools [15]. Human children are avid explorers of their material world: from four months of age infants from Westernized and traditional hunter–gatherer societies display prolonged interest in objects [16–19], and by nine months will reach for objects and begin to use them as tools [20]. Over the first year, infants actively learn about material properties, such as flexibility, rigidity and pliability [18,21,22]. Chimpanzees and orangutans display similar interests in exploring novel objects in infancy [23–26], although less combination of multiple objects occurs with nonhuman great apes [27], and sustained attention is longest in human infants [28]. These explorative behaviours enable great ape species to develop increasingly accurate motor skills in grasping, handling and manoeuvring small objects, a necessary prerequisite to tool use [29,30].

In early childhood, children spend a large amount of time engaged in object manipulation, relational play and construction [16,31–34] as they develop an understanding of the causal links between object action and environmental change [1,17,35–37]. On tests of general physical cognition examining spatial reasoning and causal understanding as well as on tool use tasks, chimpanzees and children perform similarly, or children outperform chimpanzees ([24,38]; see [39,40], for accounts of higher performance in children). It is thus thought that both species share the broader cognitive skills necessary for the mental representation of objects as tools [24,38]. However, while great apes appear to focus on change-related outcomes when observing tool use in others, human children possess key cognitive capacities for understanding intentionality which allow them to interpret and imitate the means (actions) as well as the outcomes (results) of observed tool using behaviours [2]. Children also demonstrate greater aptitudes in overcoming learned solutions to use novel, more efficient strategies than do chimpanzees, which might suggest a higher flexibility in general problem solving [41–44].

What is persistently missing from investigations into the development of human tool use, however, is representation from the large constituent of children living outside of Westernized,[3] industrialized societies. This is a pervasive problem across developmental psychology, with the majority of studies sampling exclusively from Westernized, educated, industrialized, rich and democratic (WEIRD) societies [47,48]. Such populations have been shown previously to score on the extremes of human performance, often performing differently to populations from other cultural milieus on a range of cognitive dimensions, including visual perception, analytic reasoning and working memory [47]. Similarly, the developmental landscape of WEIRD children commonly features more teaching, more formalized education and less observational and trial-and-error based learning than those growing up in small-scale or traditional hunter–gatherer communities, which may facilitate individual learning [31,33,48–50]. This suggests that the behaviour of children from WEIRD backgrounds may be culturally specific and not reflective of typical trajectories of learning [47].

To gain insight into how cultural factors influence the development of human cognitive behaviours, including tool use, a broad array of human cultural populations must be examined [51]. Calls to incorporate comparative, developmental and cross-cultural approaches within psychology are also gaining traction [52,53]. If we only use a small subset of the world's children to describe human development and cognition, how can we accurately understand the variability and breadth of human behaviour, how it evolved, and how factors like culture influence it? Furthermore, how can we extrapolate how such behaviour might differ to that of other species?

The aim of the current study was, therefore, to examine whether spontaneous tool use abilities might be generalized across children from varying cultural milieus by applying the tasks of Reindl *et al.* [9] to children growing up in transitional[4] hunter–gatherer, small-scale communities, as well as another Westernized city. We examined the tool use invention abilities of Bushmen[5] children aged 2–5 years-old

---

[3]We class Westernized societies as societies that have adopted cultural ideologies typically associated with Europe or European colonization. Most often, these include the UK, Europe, Australia, Israel and the USA. Often, the major ethnic presence in such societies is of European descent [45], and cultural practices such as formalized education are frequently adopted [46].

[4]We use transitional here as the communities we visit are exposed to some aspects of Westernized economy, services and infrastructure, while also retaining aspects of their traditional lifestyle.

[5]The people in the communities we visited refer to themselves as Bushmen, and so we respectfully do the same. They prefer to be called by their clan names, so where appropriate we do so here. Our sample of children are from three clans—the !Xun and the Khwe, both residing at Platfontein in adjacent spaces, and the ≠Khomani, located 600 km northwest in the Kalahari (Kgalagadi [Tswana] desert). We use 'Bushmen' as a generalized term when referring to all three clans, although each differ in language, culture, ethnicity and places of origin and residence. San is a Nama word meaning 'bandit' or 'forager', imposed by politicians who use it in place of 'Bushmen'. The San accept this naming for political reasons, but view it as demeaning and do not adopt it themselves [54,55].

in the Northern Cape Province of South Africa and a new sample of Australian children of the same age from a large metropolitan city. These communities were selected because they differ on several cultural features, including their level of parental pedagogical engagement (which might impact individual learning propensities), their broader lifestyle (transitional subsistence versus Western industrial) and are populations some of the authors have collaborated with for over 10 years.

If spontaneous tool invention exists within the general cognitive repertoire of humans (their zone of latent solutions), then when examining children culturally naive to great ape foraging problems, such as tool using extraction behaviours, at least some children in these cultures should spontaneously invent these behaviours. Similarly, if we share the same physical cognition capacities required for tool invention with our closest living relatives, then children in both cultures should invent tool behaviours frequently seen in wild ape populations at higher rates than those less frequently seen, reflecting shared levels of difficulty in executing the cognitive and motor demands of the tasks. Finally, if the cultural environment plays a role in the development of tool use behaviours, we may see differences in the performance trajectory or success rates of tool invention across cultures. Given that little to no research has examined spontaneous tool using behaviours cross-culturally, we provide no directional hypotheses regarding cultural differences. Rather, we aim to document any similarities and/or variation existing between distinct cultural settings.

# 2. Method

## 2.1. Participants

In total, 74 Bushmen (!Xun, Khwe, ≠Khomani) children (43 males, 31 females) from South Africa and 77 Australian children (43 males, 34 females) participated in the study, with a total of 151 children and 563 observations made across 12 tasks across cultural groups. An additional nine children were excluded due to excessive shyness (7 Bushmen; 2 Western). Children were aged from 2 to 5 years, although age in number of months was not collected in either sample, as this information is often not known in Bushmen communities (see electronic supplementary material). This age group was selected in order to replicate the age range of 2- to 3-year-olds used in Reindl *et al.* [9], with inclusion of an upper limit of age 5 to track potential age effects on task performance, as no ceiling effects were observed in the original study. All children who wished to be tested at the South African field sites were given the opportunity to do so, which led to variation in total cell sizes for each age group across samples. Due to limited access to very young children in the South African communities, only five 2-year-olds participated in the study (two in South Africa and three in Australia). The Western sample was age-matched to represent the South African sample (see electronic supplementary material, table S1).

Ethical approval was obtained from the Australian university affiliated with the authors and permission granted in the South African Bushmen communities by individuals, crèches and schools before testing commenced.[6] A community member fluent in the local language of each field site and well known to each community was recruited to translate task instructions during testing. Each translator back-translated the instructions to another community member prior to testing to ensure accurate translation.

---

[6]All paperwork was completed and sent to the South African San Council (SASC). The SASC is the 'national body of *all* [emphasis added] the San people in South Africa… It is the political body of the San'. The initial committee was elected in 2001, with subsequent elections in 2003 and 2006. The next election was not until 2018 (Andries Steenkamp 2007, personal communication). The SASC professes to protect San indigenous knowledge and became gatekeepers to San communities in regards to media and research interactions between 2007 and 2009. In March 2017, the SASC produced the San Code of Ethics (see http://www.globalcodeofconduct.org/affiliated-codes/). Their representation is not supported by all community members and is outwardly questioned by some (for a critique on the SASC's representativeness, see Parkington *et al.* [56]). The SASC do not have a website, so can be difficult to locate, and when researchers do manage to contact them, the SASC often fail to respond. The Bushman Council, established in 2009, is distinct from the SASC (although they are often confused). It is comprised of elected ≠Khomani personnel who were responsible for decisions pertaining to ≠Khomani-owned traditional farms and ≠Khomani-owned land in the Kgalagadi Transfronteir Park (Dirk Pienaar 2019, personal communication). In regard to the current research, appropriate contractual paperwork was submitted to the SASC but no response was received. We then contacted the Bushman Council office to notify them of our proposed research. Access to the communities was then facilitated through the Bushmen Council office following their approval and our SASC paperwork shown to the community leaders and onsite gatekeepers on arrival.

### 2.1.1. Bushmen sample

The samples of Bushmen children were recruited from two geographical study sites in South Africa: Platfontein and the Kalahari, and consist of three cultural groups. The Platfontein Bushmen consist of the !Xun and Khwe, who originated from Angola and Namibia respectively, but who have been living in the Northern Cape, South Africa, since the early 1990s [57]. The ≠Khomani live in the Kalahari and are South Africa's only remaining Bushmen originating from the area. Before the formalization of colonial borders, however, the ≠Khomani traversed the area into current-day Namibia and Botswana. Although all three communities were historically hunter–gatherers, none of them rely on this lifestyle for their livelihoods today: instead, they are exposed to and participate in both traditional and modern society, and experience many of the social and economic disadvantages associated with living between both [55]. Members of each community have endured a trying history of displacement, neglect, violence, and loss of cultural practices as a result of colonialism and the institution of apartheid in South Africa [54,58,59].

### 2.1.2. Platfontein sample

The final sample was 32 children recruited within crèches and kindergartens within the !Xun or Khwe communities (21 males, 11 females) of Platfontein. Platfontein is a settlement located on the outskirts of Kimberley, the capital city of the Northern Cape Province in South Africa. The township consists of basic concrete housing and self-built structures of brick and corrugated iron. Poverty, malnourishment and unemployment are high. Six per cent of adults receive their senior high school certificate here [60]. Children may attend community crèches and government-funded schools, although attendance is inconsistent. Westernized influence is moderate, and although government housing has electricity access and running water, few families can afford electricity or have a TV. Some adults do have cell phones and Internet access. In the crèches and schools, children are exposed to some Western toys such as blocks and dolls, and often play football outside, but are nevertheless relatively isolated from urban South Africa with its increasingly hybridized cultural geographies that intersect African and Western cultures [57,59].

### 2.1.3. Kalahari sample

The final sample of 42 ≠Khomani children (22 males, 20 females) was recruited from kindergartens, schools and one community centre situated on two farms 60 km south of the Kgalagadi Transfrontier Park (KTP) in the southern Kalahari. The ≠Khomani community was awarded this land in 1999 following a restitution land claim [54,57]. The ≠Khomani settlements consist of a few former farmhouses and self-built zinc and grass huts, although there is no running water, electricity or sanitation. There are government-/private-funded crèches and schools within the community, which employ local teachers and assistants; however, regular attendance is low. Children are exposed to commercial books and toys at school, and have a few of their own: they often make their own play toys for use outside, such as footballs constructed from grass, paperbark and plastic bags [55,58,59].

### 2.1.4. Australian sample

The sample of Australian children was recruited in the foyer of a science museum in Brisbane, Australia's third-largest city, or via a database of interested parents at the university laboratory. While specific demographic information was not collected for this sample, previous unpublished data collected at the science museum and laboratory indicates that the majority of participating families are from middle-class socioeconomic backgrounds and identify as Caucasian, although multiple ethnicities and economic backgrounds are represented. Young children attend preschool at age 4 in Australia, and curriculum-based schooling is compulsory until 15 years of age [61]. As a Westernized city, children living here are commonly exposed to TV, social media, commercial toys and industrial tools on a daily basis.

## 2.2. Materials

Materials were taken from Reindl et al.'s [9] Great Ape Tool Test Battery (GATTeB), consisting of 12 (child-friendly) novel extractive problem-solving tasks constructed to represent selected cultural tool behaviours observed in wild great ape populations ([11,62,63]; table 1). These tasks were split into

**Table 1.** The GATTeB tasks and testing durations.

| task (frequency) | great ape behaviour | description of task | testing time (min)[a] | criterion per community |
|---|---|---|---|---|
| insect pound (low) | use stick to pound base to break and retrieve insects | use stick to retrieve Play Doh® balls from tube by prodding them | 2 | > 1 child |
| perforate (low) | use stick to make probing holes in termite nests | use stick to perforate barrier in box to retrieve sticker below | 2 | > 1 child |
| nuthammer (low)[b] | use piece of wood/stone as hammer to crack nuts | use clay stone to crack plaster nut to retrieve sticker inside | 2 | 1 child[c] |
| algae scoop (low) | use twig to scoop for algae on water surface | use stick to scoop up strip of plastic in box of polystyrene balls | 2 | > 1 child |
| ground puncture (low) | use broad stick to puncture underground insect nest | use broad stick to puncture layer of Plasticine in box to retrieve sticker below | 3 | > 1 child |
| seed extract (low) | use twig to extract seeds from fruit/nuts | use stick to retrieve pom poms from slit in box | 2 | > 1 child |
| marrow pick (high) | use small stick to retrieve marrow from long bones | use stick to retrieve sponge with sticker attached from test tube | 1 | > 1 child |
| fluid dip (high) | use sticks to fish for honey or water | use stick to dip for paint in test tube | 1 | > 1 child |
| ant-dip-wipe (high) | use probing sticks to collect ants, then wipe off and eat | use wet stick to probe and collect polystyrene balls, then wipe into box | 3 | > 1 child |
| termite-fish leaf-midrib (high) | use stick after detaching leaf attached midway to retrieve termites from nest | subtract paper leaf from a stick through force or deliberate action then use stick with Velcro® to retrieve scourer pieces from box | 2 | > 1 child |
| lever open (high) | use stick as a lever to open up nest entrance in log or ground | use stick as lever to enlarge hole or lift Plasticine lid off mug to retrieve marble inside | 1 | > 1 child |

(*Continued.*)

| task (frequency) | great ape behaviour | description of task | testing time (min)[a] | criterion per community |
|---|---|---|---|---|
| termite fish (high) | use stick to extract termites from nest | use stick with Velcro® ends to retrieve scourer pieces with stickers from box | 1 | > 1 child |

[a]Testing durations were determined based on pilot testing by Reindl *et al.* [9] that monitored the average length of time taken for children to solve the task.

[b]This task differed from the original study. Instead of a plastic nut, a plaster nut was used that children could not tear open with their hands.

[c]Due to the complexity of this type of tool use, we follow the criterion of Bandini & Tennie [12] that correct execution of this behaviour in one child per community is sufficient to consider it above chance likelihood in each.

high- and low-frequency tasks, according to their reported occurrence in wild great ape populations. Each task was a puzzle box that contained a reward inside (such as a sticker or marble) that needed to be extracted by the child.

In this task battery, it is required that at least two children invent and correctly use the candidate tool using behaviour on each task in order for it to be considered within the individual learning capabilities of human children [9]. Because the actions required in these tasks are relatively simple tool using behaviours, two independent inventions must occur to conclude that the behaviour is unlikely to have occurred by chance. The exception is the low-frequency task of nut hammering (table 1). This is considered a complex tool using behaviour, as it requires the sequencing of multiple steps and the coordination of several objects in unison (a nut, a hammer, and an anvil) to ensure success. In the wild, this technique takes a lot longer than other simple tool behaviours to develop—both chimpanzee groups and children from hunter–gatherer communities in the Republic of Congo take several years to develop the technique [64–69]. For our *nuthammer* task, we follow the criterion recommended by Bandini & Tennie [8,12] for complex tool using behaviours: observing the behaviour in at least one individual in each cultural community is sufficient evidence for individual learning in each, as it is very unlikely that the correct sequence of actions required would arise by chance.

## 2.3. Procedure

Before testing commenced, the study was explained to parents, teachers or a community leader, and written or verbal consent obtained. All children were tested on a mat on the floor out of view of other children. The child was seated across from the experimenter (and translator, for Bushmen children) either in a science museum foyer or university laboratory for the Australian children, or outside a school, community space or crèche building for the Bushmen children. Children were first invited to play a warm-up game with the experimenter where they broke paddle pop sticks in half to construct a fence around a 'horse's paddock'. The purpose of this game was to warm up the child to the experimenter and testing set-up, and to indicate that it was okay to alter the state of materials within the games. To avoid fatigue effects, each child was given a randomized selection of four of the tasks from the GATTeB battery rather than all 12. This could be any combination of low- and high-frequency tasks.[7] The experimenter placed down a puzzle box and an appropriate tool side by side in front of the child simultaneously, without drawing direct attention to either object. The child was then invited to retrieve a reward from the puzzle box (see electronic supplementary material for instructions for each task). Each puzzle box had a time limit of 1, 2 or 3 min (table 1) in which the child could attempt to solve the task without aid. If children did not retrieve the sticker from the

[7]For example, one child might receive three low-frequency tasks and one high-frequency task, and another might receive no low-frequency tasks and four high-frequency tasks. This differed from the procedure of Reindl *et al.* [9] in which each child received two low-frequency and two high-frequency tasks, and was adopted to allow for efficiency of administration when working in the field conditions of South Africa, where re-baiting of the apparatuses was not always possible between each child's testing session.

puzzle box in the allotted time, the experimenter retrieved the sticker with the tool, and handed it to the child.[8] Children were rewarded with a small gift for participating. In the Australian sample, parents were debriefed on the task following their child's completion, and in the South African communities the kindergarten and crèche teachers as well as community gatekeepers were verbally debriefed at the conclusion of data collection.

## 2.4. Coding

Dependent variables were coded for in each of the 12 tasks as follows: (i) whether the tool was picked up by the participant (yes/no); (ii) whether the stick was used in the correct manner (yes/no; 'correct' being the typical technique used within great ape populations); (iii) whether the sticker was successfully retrieved from the puzzle box by using the tool in the correct manner (yes/no); and (iv) whether the sticker was successfully retrieved from the puzzle box using an incorrect or alternative method (yes/no; termed 'incorrect' success). Including a measure of correct and incorrect success allowed us to examine differential rates of both tool use that replicated that used by great apes and alternative methods that used a differing technique to great apes that led to success across culture. Our dataset and code is available at the Dryad Digital Repository at https://doi.org/10.5061/dryad.x0k6djhfn [70].

Inter-rater reliability coding occurred on 20% of videos for a strong agreement rating between the two coders (Cohen's kappa = 0.89). Ratings were high for all variables (pick-up = 0.88; correct use = 0.81; correct success = 0.88; incorrect success = 0.74). Following the procedure of Bandini & Tennie [12], we calculated that to detect an effect of 10% of tool inventors within our sample with 80% power, we would need a sample above 29 children in each culture (which our sample adequately exceeds).

# 3. Results

## 3.1. Performance on individual tasks

Across our tasks, at least two children used the tool correctly in each culture, reaching the criterion used in Reindl *et al.* [9] for simple tool using behaviours. The exception was the *nuthammer* task, where at least one child successfully used the tool in each culture. This reached the recommended criterion for complex tool using behaviours [12]. These findings suggest that all selected tool using tasks are within the individual learning capabilities of the children in our cultural communities. The frequency of invention for each task in each culture is available in electronic supplementary material, tables S6 and S7.

## 3.2. Effects of age, culture and frequency

We investigated whether children's performance on each of the four dependent variables (DVs; tool pick-up, correct tool use, correct success and incorrect success) varied across communities (Australian/Bushmen[9]), the age of our participants (2-, 3-, 4- and 5-year-olds), and frequency status of the tasks (high/low), with sex (male, female) entered as a control variable, as we expected no differences across sex. We ran a series of logistic generalized linear mixed models (GLMM; [71]) with a binomial error structure (as each DV was a yes/no variable) and a logit link function in R v. 3.6.1 [72] using the glmer function of the R package lme4 [73]. We included participant ID as a random intercept, and included a random slope for frequency, which allowed each subject to vary in their response across the levels of task frequency, following recommendations to keep the random effects structure 'maximal' [74]. We present the best fitting models for each variable here, with a full breakdown of our model selection process available in the electronic supplementary material.

## 3.3. Tool pick-up

All tasks except the ant-dip-wipe task were included in tool pick-up analyses. In the ant-dip-wipe task, children were actively encouraged to take the tool out of the box, as the key foraging behaviour examined

---

[8]This differed to the procedure used in Reindl *et al.* [9]. To ensure that children were motivated to continue playing for the duration of testing, we adopted this procedure so that children would receive a sticker following the allotted testing time required for each task.

[9]We conducted a comparison of performance of children from each community in South Africa and found no significant differences in their performance, so collapsed these communities into one group for all key analyses (see electronic supplementary material).

here was the wiping down action on the tool, rather than whether they would pick it up. The final model for tool pick-up contained significant effects for culture and frequency, but no effect of sex, age or any interactions. The culture effect indicated that on average Australian children picked up the tool more often than the Bushmen children, $\chi_1^2 = 64.29$, $p = <0.001$. Australian children picked up the tool in 92% of all tasks, and Bushmen children picked it up across 66% of all tasks (see electronic supplementary material for tables), showing that Australian children were 7.09 times more likely (95% CI [4.14, 12.12]) to pick up the tool than Bushmen children. The frequency effect indicated that children in both cultures were 1.68 times (95% CI [1.04, 2.70]) more likely to pick up the tool on the high-frequency tasks than the low-frequency tasks, $\chi_1^2 = 4.64$ , $p = 0.031$. Children across both cultures picked up the tool 85% of the time on high-frequency tasks but 74% of the time on low-frequency tasks.

## 3.4. Correct use

The final model for correct tool use contained significant effects for culture and frequency, but no effect of sex, age or any interactions. The culture effect indicated that on average Australian children used the tool correctly more often than the Bushmen children, $\chi_1^2 = 59.33$, $p < 0.001$. Australian children used the tool correctly 83% of the time overall, and Bushmen children used it correctly 53% of the time, revealing that Western children were 4.33 times more likely to use the tool correctly (95% CI [2.89, 6.49]; see electronic supplementary material for tables). The frequency effect indicated that children in both cultures were 1.62 times (95% CI [1.03, 2.39]) more likely to use the tool correctly on the high-frequency tasks than the low-frequency tasks, $\chi_1^2 = 6.00$, $p = 0.014$. Children across both cultures used the tool correctly 74% of the time on high-frequency tasks, but 64% of the time on low-frequency tasks.

## 3.5. Correct success

The final model for correct success on the task contained significant effects for age, culture and frequency, but no effect of sex or any interactions. The age effect indicated that in both cultures older children succeeded on the tasks correctly more often than younger children, $\chi_1^2 = 20.46$, $p < 0.001$. With every yearly increase in age, children were 1.61 times (95% CI [1.30, 1.98]) more likely to succeed on the task. The culture effect indicated that on average Australian children successfully used the tool correctly more times than the Bushmen children, $\chi_1^2 = 37.94$, $p = <0.001$, with Australian children succeeding 58% of the time, and Bushmen children succeeding 32% of the time. Australian children were 3.03 times (95% CI [2.11, 4.34]) more likely to succeed on the task in the correct manner than Bushmen children. The Australian sample also solved the tasks faster on average than our Bushmen sample (Australia: 60 s; South Africa: 74 s, $p < 0.001$). The frequency effect indicated that children in both cultures were 1.81 times (95% CI [1.27, 2.59]) more likely to succeed on the task using the tool correctly on the high-frequency tasks than the low-frequency tasks, $\chi_1^2 = 10.69$, $p < 0.001$. Children across both cultures succeeded on the tasks 53% of the time on high-frequency tasks but 38% of the time on low-frequency tasks. Figure 1 reveals the percentage of correct success for low- and high-frequency tasks in both cultural groups.

## 3.6. Incorrect success

Thirteen per cent of Western children and 10% of Bushmen children used alternative methods, such as using their hands or dissembling the boxes using force, to solve the task (see electronic supplementary material). The final model for incorrect success on the task contained a significant effect of frequency, but no effects of age, culture, sex or any interactions. Children in all communities were more likely to use alternative methods to solve low-frequency tasks than high-frequency tasks. This is most likely because the routes for success using the tool were most opaque or difficult in low-frequency tasks. This indicated that age, culture or sex did not influence the number of children succeeding on the task using alternative methods.

# 4. Discussion

The aim of the current study was to investigate whether human children from differing cultural settings could spontaneously invent novel tool using behaviours independently. Our study replicates and extends the results of Reindl et al. [9], demonstrating for the first time that young children from transitional

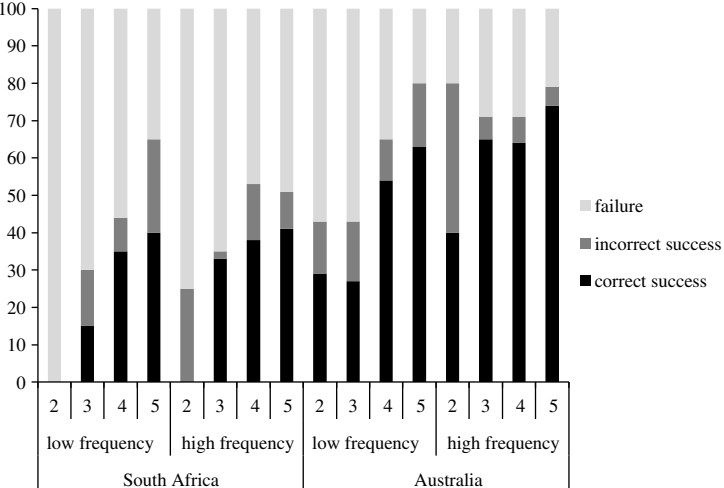

**Figure 1.** The percentage of children in each age group that succeeded in retrieving the reward correctly and incorrectly, split by cultural group and frequency.

hunter–gatherer, small-scale communities and a previously untested Westernized city, are capable of inventing all 12 selected tool use behaviours without observing or having been taught these behaviours previously.[10] This provides support for the notion that generalized tool use invention capability can occur in the absence of direct social learning mechanisms such as observation or teaching, and may be universal across human populations. It suggests that the candidate tool using behaviours examined may lie within the general physical and cognitive capabilities, or zone of latent solutions, of the human species.

Our findings show that from 2 to 3 years of age children are capable of inventing tool use behaviours and applying them correctly on their own. Across both cultural groups, children at all ages were equally likely to *use* a tool in the correct way—using the tool 74% of the time on high-frequency tasks and 64% of the time on low-frequency tasks. This closely replicates the rates of invention observed in children from another cultural group [9], and supports previous research demonstrating that from a young age children are adept tool users [29,30,36,75], and typically perform similarly to great apes on tests of physical cognition [24,38]. This suggests an inbuilt capacity for recognizing and using tools that may establish within children at 2–3 years of age.

However, as in Reindl *et al.* [9], the ability of children to *succeed* on the task improved with age, and did so across both cultures—older children retrieved the sticker using a tool far more often than younger children. This suggests that while very young children may possess the cognitive capacity to recognize the value of a tool and apply it correctly, their physical capabilities in executing the tool using behaviour may still be developing. Fine motor control is refined throughout childhood as children gain experience manipulating tools, and is necessary for accurate application in tool using tasks [29,30,76]. Similar improvement occurs in juvenile primates as they gain experience using tools and their physical strength and coordination improves [27,77].

We also found differences in the levels of tool use invention between high- and low-frequency tasks. Children at any age across culture more frequently invented, used[11] and succeeded with tool behaviours more commonly observed and reported in wild ape populations when compared with those less frequently observed. These findings suggest that certain tool using behaviours are easier to invent (for all great apes, including humans), probably because the affordances of the problem or tool are more apparent to these species [17,76], and to execute, probably because the sequence of motor actions are simpler [64]. The high-frequency tasks and tools often had clearly visible affordances, where the tool function could easily be identified. For example, in *lever open*, the lever tool was clearly tapered so

---

[10]Nut hammering was the most complicated tool behavior in our task set, and was solved only by a few of our oldest children (4–5-year-olds; see electronic supplementary materials). From the current results, it is difficult to conclude whether its invention by these children evidences it as a latent solution (which nevertheless requires more cognitive maturation than the other tool behaviours due to its complex nature), or whether its invention occurred in these few children due to unique social exposure to nut hammering within their prior experience. We suggest interpreting the results of this behavior with caution.

[11]Reindl *et al.* [9] found no frequency effect for correct tool use, see electronic supplementary material for our proposed explanations for the observed differences.

that it would easily fit the hole placed visibly at the top of the container. By contrast, low-frequency tasks often had more opaque affordances, where additional information needed to be gained from the task through exploration or prior experience—such as learning that the Plasticine cover in the ground puncture task was malleable and could be perforated to gain access to a hollow cavity beneath. Similarly, using a tool to lever a platform open (a high-frequency behaviour) requires less motor sequencing than arranging a nut, hammer and anvil correctly in order to crack a nut (a low-frequency behaviour).

Our results also indicate variation in the rates of children's tool invention on the GATTeB between cultures. Although each cultural community was capable of inventing all available tool use behaviours and executing them correctly, children from a Westernized city in Australia invented and executed the tool use behaviours correctly at significantly higher rates. This suggests that some aspect(s) of these children's cultural or social environment prior to testing served to either facilitate (in the case of Australian children) or dampen (in the case of Bushman children) their level of spontaneous tool use on the tasks, or that something about these tasks made them easier/more difficult/more enticing to solve for one group than another. This raises calls for the GATTeB to continue to be evaluated in more culturally diverse groups to identify what factors might scaffold or inhibit performance.

In the absence of additional testing, we propose potential reasons for why the reported differences arose. Even in the absence of direct social information like teaching and observation, the mere presence of artefacts made by others might support individual tool-based learning indirectly, and shape it towards those forms and techniques already employed within a cultural group. In wild non-human primate populations, discarded materials and tools used previously by others may aid in infants' and juveniles' explorative tool use, even in the absence of a social model [78]. In humans, tool templates made by another person aid children in creating their own tools in experimental conditions [79] and in everyday community life [31,51,80]. It is likely then that the materials we employed in our task battery might have transmitted indirect cultural information, which might have facilitated performance within the Australian children.

Although efforts were made to ensure tasks were comparably novel to each community, they were made from materials commonly found in Westernized households and preschools—including polystyrene balls, cardboard, Plasticine and clay. These materials were, on observation, far less prevalent in the households or schools of the Bushman communities. Australian children may have benefitted from seeing these materials and their uses before in their homes and schools, making their presenting affordances within the tasks and tools more salient or recognizable. The fact that our Australian sample solved the tasks faster on average than our Bushmen sample lends support to this hypothesis. For example, in *ground puncture*, knowing that Plasticine is malleable aids in solving the task—children who have yet to encounter this material would take more time to discover that the Plasticine roof could be punctured to retrieve the sticker beneath. Future research could provide Bushmen children with similar materials for several weeks prior to testing, in order to examine how prior experience with such materials might facilitate their tool invention rates. Alternatively, previous research has found that adapting Westernized tasks to employ local materials can bridge the gap in performance between participants from Westernized and non-Westernized cultural backgrounds [81,82]. Developing a comparable task set locally would be a valuable step for future research within transitional small-scale societies.

It is worth noting that the children in our samples were on average older than the children in Reindl *et al.*'s study [9], meaning that greater cultural exposure prior to testing may have cemented children's tool use behaviours. Perhaps restricting our sample to a younger, more culturally 'naive' cohort could have reduced the group differences found. However, if we look at the performance seen between our youngest participants (2–3-year-olds), these differences are still apparent, and we found no interactions between cultural influence and age across our sample. This suggests that any indirect effects of cultural influence, previous social learning opportunities and/or material familiarity on spontaneous tool invention and associated cognitive abilities must emerge earlier than this age in children. The employment of alternative strategies to solve the puzzle boxes, such as using hands/ fingers, applying brute force or disassembling, cannot explain our cultural differences, as an equally small percentage of children in each culture solved the tasks using such methods. Rather, it appears that the sociocultural niche of a child's environment, or the artefacts in it, might influence their perception of tool-like objects from an early age, and in doing so affect their potential to pick up a tool and use it in a certain way to solve a problem [78,80].

It must also be noted that major differences exist in the economic backgrounds of our communities, which could have also generated differences in performance. Our Bushmen communities live within the rural poor of sub-Saharan Africa, one of the most poverty-stricken areas globally, typified by remote locations that make public services and livelihood opportunities limited [54,83]. It is possible that some of the Bushmen children we tested had not eaten the morning of testing, which could have affected their cognition and performance. Our Australian sample, by contrast, was from a relatively wealthy city and can be assumed to have consistent access to public and health services. We cannot rule out that differences in the nutrition or socioeconomic opportunities of these communities may have driven the differences we found in tool invention performance. This is a clear limitation of the current study, and future replications of this study could include a low socioeconomic sample from Australian rural areas, and a middle-income sample from a metropolitan city in South Africa, to further investigate and isolate the role nutritional factors might exert on children's performance on the tasks, contrasting them to the impact of other cultural factors, such as education level or pedagogical framework.

Despite these differences, the current research demonstrates that children from diverse cultural settings and economic circumstances can independently invent the same 12 tool using behaviours observed in wild great ape populations. This supports the notion that humans and other great ape species may share similar underlying physical cognitive capacities that allow for flexible tool use invention [24,38]. These may establish in children at 2–3 years of age and be individually inventible even in the absence of direct social influences like teaching and observation. Being able to recognize and use tools to solve problems is a highly adaptive skill, enabling access to a greater range of food sources and foraging opportunities than would otherwise be available. It is likely that these tool invention capacities already existed within the last common ancestor of all great apes, some 14 Ma [9,13]. This would suggest the existence of an (unobservable) organic tool age that preceded the stone age observed within primate technological history.

Nevertheless, the complexity and diversity of tool kits extant in human communities around the world today cannot solely be the work of individual invention. Additional processes are necessary to support these cumulative technologies: including sophisticated mechanisms of social learning such as imitation and teaching [2,6], combined with the continual improvement of inventions by others [84,85] or the selective retention of copying errors [4]. While young children may be shaped very early on to perceive and use tools in the established ways of their group, within the years of adolescence and adulthood, a shift from social learning to more independent thinking may enable them to contribute their own new inventions into a group's repertoire, and further advance the material culture of their group [51,86].

Ethics. This study and its data collection methods and processes were reviewed and approved by the University of Queensland's Behavioural and Social Sciences Ethical Review committee (BSSERC; approval no. 2014001272). Permission to collect the data was provided by the Bushmen Council and appropriate paperwork shown to the community leaders and onsite gatekeepers on arrival. Informed consent was obtained through written or verbal consent from children and their guardians before testing commenced.

Data accessibility. Data are available at Dryad Digital repository: https://doi.org/10.5061/dryad.x0k6djhfn [70].

Authors' contributions. K.N. adapted the design for cross-cultural testing, collected and analysed the data, interpreted the results and wrote the manuscript. E.R. designed the study, built study materials, assisted in the analysis of the data and interpretation of results and critically revised the manuscript. C.T. designed the study and critically revised the manuscript. J.G. and K.T. facilitated data collection in South Africa and critically reviewed the manuscript. M.N. adapted the design for cross-cultural testing, facilitated data collection and critically reviewed the manuscript. All authors provided final approval for publication.

Competing interests. The authors declare no competing interests.

Funding. This study was supported by an Australian Research Council Discovery Project grant no. (DP140101410) awarded to M.N. At the time of writing, C.T. has received funding from the European Research Council (ERC) under the European Union's Horizon 2020 research and innovation programme (grant agreement no. 714658; STONECULT project).

Acknowledgements. We thank the Queensland Museum's staff and patrons for the assistance in data collection. We thank the staff and parents at the University of Queensland's Early Cognitive Development Centre for their assistance in data collection. We thank the parents and children of the !Xun, Khwe and ≠Khomani communities for their generous assistance with data collection. We thank our translators, Jason and Deon, for their invaluable assistance in translating the tasks in Platfontein and the Kalahari, respectively. Our thanks to Caritas for permission to work at its early childhood centres in Platfontein. We thank Marissa Wu and Daniel Chen for their help in data coding for this project. We thank Bruce Rawlings for his helpful comments on an earlier version of this manuscript. We thank Sean Murphy for his statistical advice.

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
