## [Reviewer comments · Royal Society Open Science]

Review History

RSOS-192240.R0 (Original submission)

Review form: Reviewer 1

Is the manuscript scientifically sound in its present form?

Yes

Are the interpretations and conclusions justified by the results?

Yes

Is the language acceptable?

Yes

Do you have any ethical concerns with this paper?

No

Have you any concerns about statistical analyses in this paper?

No

Recommendation?

Accept with minor revision (please list in comments)

Comments to the Author(s)

We found the manuscript is direct, concise, and easy to read and we believe that it ultimately deserves publication.

Nevertheless, we did have a few comments about it:

It would be helpful to a broad readership if you could further elaborate why identifying a cognitive trait such as the capacity for certain tool innovations in adult animals versus juvenile humans would suggest the presence of that trait in a common ancestor? Moreover, do you mean the common ancestor of all great apes or the common ancestor of humans and chimps?

It is repeated along the manuscript that “culture” is the origin of the differences found between the Australian and Bushmen children. However, culture is commonly used to describe learned behaviours and knowledge, but there are possible origins of these differences that fall away of this concept (impoverishment, and its results, is probably the most evident). Since the authors can only speculate about the origin of these differences, perhaps a broad term would be more appropriate.

In line 135 it is stated that “in the South African communities, very few two-year-old children were tested”, but no numbers are given to the reader. This is important since no two-year-old children succeed in the South African communities neither in low nor in high frequency tasks. In the Australian sample, however they solved the low frequency task in a comparable way to three-year-old children, and high frequency tasks (combining correct and incorrect solving) virtually as five years-old children.

Although required mechanical procedures of the GATTeB tasks are clearly similar to the ones that great apes face in the wild, there are probably big differences in the innovative strength required for solving them. Few of them give clear clues about what to do (eg. stick with Velcro; generally, two combinable objects close by, and easily to recognizable from the environment) and are probably far away from the real ill-structured problem that a hypothetical ape would face in nature. It may be important to discuss these limitations when presenting the tasks.

In line 330, it is stated that with every yearly increase in age, the children of both populations were 1.61 times (95% CI [1.30, 1.98]) more likely to succeed on the task. If the difference between both groups is cultural, as is stated in the manuscript, should not we see an increase in these differences with time and cultural exposure?

This was a very interesting read and we are looking forward to see a revision as well as the results of future adaptations of the GATTe

Review form: Reviewer 2

Is the manuscript scientifically sound in its present form?

Yes

Are the interpretations and conclusions justified by the results?

Yes

Is the language acceptable?

Yes

Do you have any ethical concerns with this paper?

No

Have you any concerns about statistical analyses in this paper?

No

Recommendation?

Accept with minor revision (please list in comments)

Comments to the Author(s)

See attached (Appendix A).

Review form: Reviewer 3 (Annette Hohenberger)

Is the manuscript scientifically sound in its present form?

No

Are the interpretations and conclusions justified by the results?

No

Is the language acceptable?

Yes

Do you have any ethical concerns with this paper?

No

Have you any concerns about statistical analyses in this paper?

No

Recommendation?

Major revision is needed (please make suggestions in comments)

Comments to the Author(s)

See attached file (Appendix B).

Decision letter (RSOS-192240.R0)

10-Mar-2020

Dear Dr Neldner

On behalf of the Editors, I am pleased to inform you that your Manuscript RSOS-192240 entitled "A cross-cultural investigation of young children's spontaneous invention of tool use behaviors" has been accepted for publication in Royal Society Open Science subject to minor revision in accordance with the referee suggestions. Please find the referees' comments at the end of this email.

The reviewers and handling editors have recommended publication, but also suggest some minor revisions to your manuscript. Therefore, I invite you to respond to the comments and revise your manuscript.

- Ethics statement

If your study uses humans or animals please include details of the ethical approval received, including the name of the committee that granted approval. For human studies please also detail

whether informed consent was obtained. For field studies on animals please include details of all permissions, licences and/or approvals granted to carry out the fieldwork.

- Data accessibility

If you wish to submit your supporting data or code to Dryad (<http://datadryad.org/>), or modify your current submission to dryad, please use the following link:
<http://datadryad.org/submit?journalID=RSOS&manu=RSOS-192240>

- Competing interests

- Authors' contributions

- Acknowledgements

- Funding statement

Because the schedule for publication is very tight, it is a condition of publication that you submit the revised version of your manuscript before 19-Mar-2020. Please note that the revision deadline will expire at 00.00am on this date. If you do not think you will be able to meet this date please let me know immediately.

If your manuscript is newly submitted and subsequently accepted for publication, you will be asked to pay the article processing charge, unless you request a waiver and this is approved by Royal Society Publishing. You can find out more about the charges at <https://royalsocietypublishing.org/rsos/charges>. Should you have any queries, please contact opscience@royalsociety.org.

Once again, thank you for submitting your manuscript to Royal Society Open Science and I look

forward to receiving your revision. If you have any questions at all, please do not hesitate to get in touch.

on behalf of Dr Teodora Gliga (Associate Editor) and Essi Viding (Subject Editor)
 openscience@royalsociety.org

Associate Editor Comments to Author (Dr Teodora Gliga):

I have now received reviews from 3 experts in the field, all of which commented on the originality of the study and the clarity of the manuscript. There are however a few issue that need further clarification or commenting on (see reviews). In particular, i urge you to comment on the possibility that your findings are not explained by cultural differences but by other cross-cultural factors, linked to SES, and that may have interfered with performance.

Reviewer comments to Author:

Reviewer: 1

Comments to the Author(s)

We found the manuscript is direct, concise, and easy to read and we believe that it ultimately deserves publication.

Nevertheless, we did have a few comments about it:

It would be helpful to a broad readership if you could further elaborate why identifying a cognitive trait such as the capacity for certain tool innovations in adult animals versus juvenile humans would suggest the presence of that trait in a common ancestor? Moreover, do you mean the common ancestor of all great apes or the common ancestor of humans and chimps?

It is repeated along the manuscript that “culture” is the origin of the differences found between the Australian and Bushmen children. However, culture is commonly used to describe learned behaviours and knowledge, but there are possible origins of these differences that fall away of this concept (impoverishment, and its results, is probably the most evident). Since the authors can only speculate about the origin of these differences, perhaps a broad term would be more appropriate.

In line 135 it is stated that “in the South African communities, very few two-year-old children were tested”, but no numbers are given to the reader. This is important since no two-year-old children succeed in the South African communities neither in low nor in high frequency tasks. In the Australian sample, however they solved the low frequency task in a comparable way to three-year-old children, and high frequency tasks (combining correct and incorrect solving) virtually as five years-old children.

Although required mechanical procedures of the GATTeB tasks are clearly similar to the ones that great apes face in the wild, there are probably big differences in the innovative strength required for solving them. Few of them give clear clues about what to do (eg. stick with Velcro; generally, two combinable objects close by, and easily to recognizable from the environment) and are probably far away from the real ill-structured problem that a hypothetical ape would face in nature. It may be important to discuss these limitations when presenting the tasks.

In line 330, it is stated that with every yearly increase in age, the children of both populations were 1.61 times (95% CI [1.30, 1.98]) more likely to succeed on the task. If the difference between

both groups is cultural, as is stated in the manuscript, should not we see an increase in these differences with time and cultural exposure?

This was a very interesting read and we are looking forward to see a revision as well as the results of future adaptations of the GATTe

Reviewer: 2

Comments to the Author(s)
See attached

Reviewer: 3

Comments to the Author(s)
See attached file

Author's Response to Decision Letter for (RSOS-192240.R0)

See Appendix C.

Decision letter (RSOS-192240.R1)

15-Apr-2020

Dear Dr Neldner,

It is a pleasure to accept your manuscript entitled "A cross-cultural investigation of young children's spontaneous invention of tool use behaviors" in its current form for publication in Royal Society Open Science. The comments of the Editors and reviewers who reviewed your manuscript are included at the foot of this letter.

Kind regards,
Lianne Parkhouse
Royal Society Open Science
openscience@royalsociety.org

on behalf of Dr Teodora Gliga (Associate Editor) and the Subject Editor
openscience@royalsociety.org

Associate Editor Comments to Author (Dr Teodora Gliga):

Thank you for being responsive to reviewers' comments and improving an already very interesting manuscript with clarifications.

Appendix A

RSOS-192240 Review

This article outlines a single study on cross-cultural comparisons of 2-5 year-old's spontaneous tool use. The study found that children from both Western, Australian and non-Western, South African Bushmen (consisting of 3 different cultures) were able to spontaneously use tools successfully to solve all 12 problems, even without any instruction or social learning. The rate of success in these tasks also corresponded with the frequency with which these tasks are observed in non-human great ape populations. The authors argue that this is evidence for a shared ability in our common ancestor with the other great apes. Additionally, they found differences in the rates at which the Western versus non-Western children used and executed tool use in these novel tasks, suggesting that culture/social environment, such as affordances of the task materials, does influence tool innovation to some extent beyond the underlying shared capacity to do so.

I found this to be an extremely well-written, clear and comprehensive article. I just have a few suggestions:

1. I think that the importance of the distinction between 'correct' and 'incorrect' innovation needs to be explained more explicitly (presumably 'incorrect' innovation is not evidence for your hypothesis that these capacities are shared with nonhuman great apes). The label is a bit misleading, perhaps call it something like 'Target Success' vs 'Alternative Success'?
2. Although personally I don't find this particularly convincing, I'd expect counter-arguments to these findings to bring up the fact that we do not know whether children have had experience with similar tools before. I could imagine that the Australian children may have seen nature programmes showing these behaviours before, or that in both cultures there may be knowledge that these behaviours exist in wild animals. This should be discussed.
3. Throughout the discussion, it is often not clear whether you are talking about age or cultural differences. For example, pg. 19 line 272 I think should read 'Across both cultures' rather than 'within', as this age comparison is collapsed across cultures. The same is true for pg. 20 line 392. It might help to add a sentence at the beginning of each section to explain that you are talking about age diffs in the 1st section and cultural diffs in the 2nd.
4. The Excel datafile would be more accessible if it included information about what the number codes stand for (E.g. Culture 0, 1 etc.).

Appendix B

“A cross-cultural investigation of young children’s spontaneous invention of tool use behaviors”

by Neldner, Reindl, Tennie, Grant, Tomaselli, & Nielsen

Review of RSOS-192240 by Annette Hohenberger

Short summary, evaluation:

This study is concerned with discerning the basic cognitive abilities present in young (2-5-year-old) children for the invention of tool use behaviors – similar to those in great apes. Going beyond WEIRD societies, the authors compare spontaneous, non-instructed tool use behaviors with a comprehensive battery of 12 pre-validated tasks (GATTeB) in a traditional culture (3 Bushmen samples in South Africa) with one not yet studied Western sample (from Brisbane, Australia). Tasks have either low or high levels of occurrence (in the ape population on which the problems are modeled). The authors find that children from both cultures are capable of solving these problems by using tools spontaneously. Success in these problems is modulated by age, frequency of the task, and culture such that older children obtain higher scores than younger ones, high-frequency tasks are easier than low-frequency tasks and Australian children obtain higher scores than South African children. The authors conclude that the basic capacity to invent appropriate tool use behaviors emerges early in childhood (which may hint at an early common ancestor of humans and the great apes), and that frequency of the behavior as well as culture influence performance.

Strengths of the study: The following aspects are novel and make the present study a valuable contribution to the field of (comparative) cognitive development:

- (1) Including hitherto neglected populations: Three traditional Bushmen samples and one new Western sample.
- (2) High *N* (151 participants from both cultures)
- (3) Comprehensive and pre-validated test battery, modeled after real-life problems for primates
- (4) Clear statistical analyses
- (5) Substantial and well-documented supportive material; online-accessible data
- (6) Well-written, coherent manuscript

Weaknesses of the study:

Despite the above virtues, I have one major conceptual and one major presentational concern which I would like the authors to address in a revision of their paper.

General (major) issues:

1. Conceptual issue (confound of culture/SES/setting): I see a big confound between three different dimensions of the populations tested: (1) Culture: Western vs traditional, (2) SES/education: rich/educated vs poor/uneducated, (3) setting: Museum vs home setting (which could also be interpreted as a specific selection of the topmost SES/educational level of the Australian sample). This confound makes it hard for me to follow the authors’ conclusion with respect to the cultural issue: they claim that cultural environment (Western vs traditional) may facilitate or hamper appropriate tool use. However, from the information they provide on the two broad samples (1 Western; 3 traditional) and the settings under which they observed their participants it is obvious that the cultural dimension covaries with SES and setting. The authors are aware of this fact but discuss it only at the end of their study, in the discussion (when it is too late). There should be a justification why these samples were drawn knowing that this confound prevailed, in the introduction. After all, it may not be the culture but poverty or the setting that brings about the difference in performance of the two groups. Children raised in poverty, with little formal education and intellectual stimulation (some even on an empty

stomach, as the authors suspect), will approach a cognitive puzzle in a very different way as children raised under safe economic conditions, with high formal education and permanent intellectual stimulation.

2. Presentational issue: The study has clear limitations, the most obvious one being the confound between culture, SES/education, and setting, mentioned above. These limitations should be stated and addressed more clearly. There are various ways of achieving this aim. Either they are discussed within the discussion section (as currently done) but then they need to be clearly stated as limitations. Alternatively, the authors may want to add a section entitled "Limitations" at the end of the discussion where they list them collectively (as is frequently done in other journals as well.) Since these limitations call for future research, yet another alternative is to combine both, e.g. in a section entitled "Limitations and future studies". In any case the authors should clearly identify these limitations and acknowledge to what extent their conclusions are affected by them and how they might be overcome in the future (which they partially did already).

Specific (minor) in-line comments:

Highlights: (here I refer to the "page" and "line")

P 3, L12: maybe specify the two populations after stating "from two diverse cultural contexts"

Main text: (here I refer to the "line" of the main text only)

7: maybe explain the "ratchet effect" more clearly and trace it back to Tomasello (1999) (and possibly to Vygotsky (1978) before mentioning subsequent literature (Boyd & Richerson, 2005, Eerkens & Lipo, 2005).

32-34 (and compare 205-207): it did not become clear to me why invention by 2 participants is considered sufficient to rule out chance. Wouldn't this criterion vary with sample size? Some more justification might be necessary here. The information provided in 267-270 seems to be relevant in this respect as well, but it is not entirely clear to me how.

43: The cognitive and physical capacities needed for tool use are, among others, addressed in the comprehensive work of Francois Osiurak. A suitable paper in this respect is:

Osiurak, F., & Badets, A. (2016). Tool use and affordance: Manipulation-based versus reasoning-based approaches. *Psychological Review*, 123, 534-568.

Although it is not developmental, this paper seems relevant since it includes aspects that are also relevant in development, namely object knowledge (as in manipulation-based approaches) and mechanical knowledge (as in reasoning-based approaches).

145-184 (on the 3 Bushmen samples): I found the characterization of the 3 Bushmen samples quite heterogeneous. It might be better to provide information about the same (selected) criteria, e.g., geographical location, history, SES, setting (of the testing), influence of Westernized society, and, most importantly, aspects relevant for tool-use (playing with traditional toys (manufactured from local materials) or with Western toys). These criteria could also be used for the comparison of the South-African sample (as a whole) with the Australian sample (185-195).

255: Were the parents of the Bushmen children also informed after the study, in some way?

267: Besides overall kappa, please provide kappa values for each of the 4 scores (a-d).

280: please indicate explicitly where in the supplementary materials this information can be found (Table S6 or S7 (?))

440: the information that Australian children could solve the tasks significantly faster on average than Bushmen children (60 s vs 74 s, respectively) should be presented before in the results section or be presented and referred to in the supplementary materials.

455: there is a left-over preposition “in” which should be deleted.

Endnotes: (here I refer to the “page” and “line”)

Endnote 4, (P 37, line 29 – p 38, line 7: As mentioned in my comments to the editor in the section on “ethics” in the online referee system, I would suggest to limit this endnote to the relevant information explaining why it was difficult to obtain consent from the South African San Council (SASC) but leave out (most of) the historical material.

Appendix C

RSOS Response to Reviewers

Thank you for your helpful comments on our manuscript submission entitled '*A cross-cultural investigation of young children's spontaneous invention of tool use behaviors*'. All reviewers raised constructive and illuminating points, including the primary concern regarding the question of alternative mechanisms that might explain the community differences found. All reviewers raised points regarding nutritional or SES differences. We acknowledge that we cannot identify or isolate how much a role these factors contributed to our findings, nor rule out that they may explain some difference between our cultural communities, rather than cultural variations. We have thus acknowledged more explicitly our lack of investigation into nutrition/SES as a shortcoming of the paper in **lines 508-511** and provided recommendations for how its role may be identified in future research in **lines 513-515**.

Lines 508-515: "We cannot rule out that differences in the nutrition or socioeconomic opportunities of these communities may have driven the differences we found in tool invention performance. This is a clear limitation of the current study, and future replications of this study could include a low socioeconomic sample from Australian rural areas, and a middle-income sample from a metropolitan city in South Africa, to further investigate and isolate the role nutritional factors might exert on children's performance on the tasks, contrasting them to the impact of other cultural factors, such as education level or pedagogical framework"

We do not think, however, that this discrepancy dampens the general finding that across such distinct differences we still find invention of all 12 tool use behaviors in both our communities. We consider this strong support for the notion that children have established capacities for spontaneous tool invention at a young age. We emphasise this in **lines 516-521**.

Lines 516-521: "Despite these differences, the current research demonstrates that children from diverse cultural settings and economic circumstances can independently invent the same 12 tool using behaviors observed in wild great ape populations. This supports the notion that humans and other great ape species may share underlying physical cognitive capacities that allow for flexible tool use invention. These may establish in children at 2-3 years of age..."

We maintain that prior sociocultural exposure to cultural artefacts or materials may be the most likely driving factor underlying our cultural differences (generating a facilitation effect in Australian children; **lines 444-450;464-470**). We retain this argument from the first version of the manuscript.

We also now provide further justification for the selection of our cultural groups, stated in **lines 119-123**. We selected these communities "because they differ in their level of parental pedagogical engagement (which might impact individual learning propensities), their broader lifestyle (transitional subsistence versus Western industrial) and are populations some of the authors have

collaborated with for many years” Our aim was to work with populations that were well known to us and differed distinctly to those populations already tested on this paradigm, and those most commonly tested in psychological research in general (ie. WEIRD populations, see Heinrich et al., 2010).

Other general concerns raised regard how we refer to the cultural differences existing between the communities. We believe that we have duly acknowledged the broadness of the term ‘culture’ by providing detail about how the communities differ socially and culturally in our introduction of the communities in **lines 119-123** (quoted in the previous paragraph), the description of their qualitative features as stated in the participants section of the Methods, and in our conservative interpretation of our findings (see lines **495-498**; “it appears that the sociocultural niche of a child’s environment, or the artefacts in it, might influence their perception of tool-like objects from an early age, and in doing so affect their potential to pick up a tool and use it in a certain way to solve a problem”). However, where possible we now refer to the ‘cultural context’ or ‘cultural setting’ (lines **17, 137, 389, 517**) of the communities, which we believe is a more conservative, and overarching, term.

We hope this revision satisfies all reviewers and editors and that the manuscript is now acceptable for publication. All changes are highlighted in yellow in the manuscript. We have responded to the specific concerns from each reviewer below:

Reviewer 1

R1: *It would be helpful to a broad readership if you could further elaborate why identifying a cognitive trait such as the capacity for certain tool innovations in adult animals versus juvenile humans would suggest the presence of that trait in a common ancestor? Moreover, do you mean the common ancestor of all great apes or the common ancestor of humans and chimps?*

This first comment queries why examining adult chimpanzees and juvenile children allows us to presume the existence of a cognitive trait in a shared common ancestor. To aid in explaining how the presence of this trait in both species suggests it emerged in a shared common ancestor, we have added explanatory sentences in **lines 20-24** and retain our explanation from the first version in **lines 525-529**.

Lines 20-24: “Understanding the extent to which young children may invent simple tool behaviors by individual learning, without the need for social direction, can also help identify when such behaviors may have emerged within our evolutionary history, and which living relatives we might share these capacities with”.

R1: *It is repeated along the manuscript that “culture” is the origin of the differences found between the Australian and Bushmen children. However, culture*

is commonly used to describe learned behaviours and knowledge, but there are possible origins of these differences that fall away of this concept (impoverishment, and its results, is probably the most evident). Since the authors can only speculate about the origin of these differences, perhaps a broad term would be more appropriate.

This question regarding cultural differences reflects the general concerns raised above. We hope our listed changes to the manuscript (see above for recognition of nutritional/SES impacts in **508-515**) adequately address this concern regarding the use of the term “culture” in the manuscript. We have broadened this to “cultural context/setting”, which may encompass broader implications.

R1: *In line 135 it is stated that “in the South African communities, very few two-year-old children were tested”, but no numbers are given to the reader. This is important since no two-year-old children succeed in the South African communities neither in low nor in high frequency tasks. In the Australian sample, however they solved the low frequency task in a comparable way to three-year-old children, and high frequency tasks (combining correct and incorrect solving) virtually as five years-old children.*

We have added the number of two year olds tested in **lines 151-153**.

R1: *Although required mechanical procedures of the GATTeB tasks are clearly similar to the ones that great apes face in the wild, there are probably big differences in the innovative strength required for solving them. Few of them give clear clues about what to do (eg. stick with Velcro; generally, two combinable objects close by, and easily to recognizable from the environment) and are probably far away from the real ill-structured problem that a hypothetical ape would face in nature. It may be important to discuss these limitations when presenting the tasks.*

We have addressed your point regarding the difference in environmental context in which apes and children might encounter these problems by adding **endnote 2**, referenced in **line 35**. We agree that the circumstances, although similar to those in which apes would find them in the wild, indeed may be more salient here for children. However, the manner in which we present the tool and problem seeks to represent how naïve apes would encounter them in populations where other group members already utilize the tool using behavior. We are interested in tracking the reinvention of the form of the behaviour itself, and not in how location or stimulus enhancement etc. mechanisms might affect them. We have explained this point in endnote 2.

R1: *In line 330, it is stated that with every yearly increase in age, the children of both populations were 1.61 times (95% CI [1.30, 1.98]) more likely to succeed on the task. If the difference between both groups is cultural, as is stated in the manuscript, should not we see an increase in these differences with time and cultural exposure?*

It is indeed interesting that we find no interaction effects between age and culture on any of our dependent variables, as one might initially predict. What we find instead is independent effects of age and cultural group. We believe this indicates that any cultural effects that impact tool-based cognition and behaviour in children are established in the earlier years of life, at 2-3 years, which is why we see no further compounding improvements or differences according to culture on this task following this age. Research indeed suggests that many of the socio-cognitive and physical abilities that lead to the adoption of tool use behaviors in children occur between 1-2 years, including inferring intentionality, imitative learning, fine motor skills etc (Tomasello, 1999; Bjorklund & Gardiner, 2011; Fontenelle et al., 2007). We retain our references to this literature from the first manuscript in **lines 51-66**, adding more detail in relation to intentional understanding in **lines 79-83**, and retain our statements that the cultural influences children were exposed to prior to testing may have exerted the differences in tool performance observed (see **lines 489-492**: “this suggests that any indirect effects of cultural influence, previous social learning opportunities and/or material familiarity on spontaneous tool invention and associated cognitive abilities must emerge earlier than this age in children”). However, it is also possible that instead we are detecting a stable group difference due to nutritional differences that persist across age groups, which also might affect cognition. We list this potential alternative explanation here **508-515**.

Reviewer 2

R2: *I think that the importance of the distinction between ‘correct’ and ‘incorrect’ innovation needs to be explained more explicitly (presumably ‘incorrect’ innovation is not evidence for your hypothesis that these capacities are shared with nonhuman great apes). The label is a bit misleading, perhaps call it something like ‘Target Success’ vs ‘Alternative Success’?*

We understand why these terms might be confusing. However, we choose to retain our terms “incorrect success” and “correct success” because these are the terms used in the original paper we are replicating (Reindl et al., 2016) and we wish to keep them for consistency. This will aid readers who wish to consult both papers in their research. However we now explain these terms more clearly when we introduce them in **lines 288-291** in the Coding section.

R2: *Although personally I don’t find this particularly convincing, I’d expect counter-arguments to these findings to bring up the fact that we do not know whether children have had experience with similar tools before. I could imagine that the Australian children may have seen nature programmes showing these behaviours before, or that in both cultures there may be knowledge that these behaviours exist in wild animals. This should be discussed.*

We believe that it is unlikely that a large number of children had prior knowledge of many of these tool using behaviours in ape populations, however we now include mention the possibility of some having prior knowledge of some

tasks in **endnote 1** cited in **line 34**. And so, while such influence cannot be fully excluded, it would fail to account for the overall effects found (and Reviewer 1 seems in agreement with us here). We believe the familiarity of the materials would exert stronger influence, which we highlight in the discussion in **lines 444-450;464-470**.

R2: *Throughout the discussion, it is often not clear whether you are talking about age or cultural differences. For example, pg. 19 line 272 I think should read ‘Across both cultures’ rather than ‘within’, as this age comparison is collapsed across cultures. The same is true for pg. 20 line 392. It might help to add a sentence at the beginning of each section to explain that you are talking about age diffs in the 1st section and cultural diffs in the 2nd.*

3. Thank you for highlighting these discrepancies. We have amended our statements in **line 401** to “across both cultural groups” and in **line 412** “did so across both cultures” and **line 423** “children at any age across culture”. We retain the remainder of our descriptions of the main effects from the first manuscript as we deem them now clear enough for clear interpretation.

R2: *The Excel datafile would be more accessible if it included information about what the number codes stand for (E.g. Culture 0, 1 etc.).*

4. We have now updated our data repository files to provide a key for our coding system and uploaded this revised version into our data repository on Dryad (available at Dryad: 10.5061/dryad.x0k6djhn).

Reviewer 3 (AH)

R3: *1. Conceptual issue (confound of culture/SES/setting): I see a big confound between three different dimensions of the populations tested: (1) Culture: Western vs traditional, (2) SES/education: rich/educated vs poor/uneducated, (3) setting: Museum vs home setting (which could also be interpreted as a specific selection of the topmost SES/educational level of the Australian sample). This confound makes it hard for me to follow the authors’ conclusion with respect to the cultural issue: they claim that cultural environment (Western vs traditional) may facilitate or hamper appropriate tool use. However, from the information they provide on the two broad samples (1 Western; 3 traditional) and the settings under which they observed their participants it is obvious that the cultural dimension covaries with SES and setting. The authors are aware of this fact but discuss it only at the end of their study, in the discussion (when it is too late). There should be a justification why these samples were drawn knowing that this confound prevailed, in the introduction. After all, it may not be the culture but poverty or the setting that brings about the difference in performance of the two groups. Children raised in poverty, with little formal education and intellectual stimulation (some even on an empty stomach, as the authors suspect), will approach a cognitive puzzle in a very different way as children raised under safe economic conditions, with high formal education and permanent intellectual stimulation.*

Thank you for raising this important point. As stated above, we have added a paragraph highlighting this potential alternate explanation for our results in **lines 508-515**:

“We cannot rule out that differences in the nutrition or socioeconomic opportunities of these communities may have driven the differences we found in tool invention performance. This is a clear limitation of the current study, and future replications of this study could include a low socioeconomic sample from Australian rural areas, and a middle-income sample from a metropolitan city in South Africa, to further investigate and isolate the role nutritional factors might exert on children’s performance on the tasks, contrasting them to the impact of other cultural factors, such as education level or pedagogical framework”

We do not think, however, that this discrepancy dampens the general finding that across such distinct differences we still find invention of all 12 tool use behaviors in both our communities. We consider this strong support for the notion that children have established capacities for spontaneous tool invention at a young age. We emphasise this in **lines 516-521**.

Lines 516-521: “Despite these differences, the current research demonstrates that children from diverse cultural settings and economic circumstances can independently invent the same 12 tool using behaviors observed in wild great ape populations. This supports the notion that humans and other great ape species may share underlying physical cognitive capacities that allow for flexible tool use invention. These may establish in children at 2-3 years of age...”

We also now provide further justification for the selection of our cultural groups despite their differences in SES, stated in **lines 119-123**. We selected these communities “because they differ in their level of parental pedagogical engagement (which might impact individual learning propensities), their broader lifestyle (transitional subsistence versus Western industrial) and are populations some of the authors have collaborated with for over 10 years”. Our aim was to work with populations that were well known to us and differed distinctly to those populations already tested on this paradigm, and those most commonly tested in psychological research in general (ie. WEIRD populations, see Heinrich et al., 2010). However, it is true that these populations often differ distinctly on other dimensions, such as SES. We make recommendations for how this factor could be measured in future studies in **lines 513-515**.

We also wish to clarify that all our children, in all communities, were tested in crèches, kindergartens, schools or museums. Therefore we think that the setting in which they were tested was fairly comparable and does not present a confound across cultural groups. We have added more specification about the testing setting for Africa groups in **lines 178-179**: “recruited within crèches and kindergartens with the !Xun or Khwe communities” and in **lines 194**: “recruited from kindergartens, schools and one community center” to clarify this.

2. Presentational issue: The study has clear limitations, the most obvious one being the confound between culture, SES/education, and setting, mentioned above. These limitations should be stated and addressed more clearly. There are various ways of achieving this aim. Either they are discussed within the discussion section (as currently done) but then they need to be clearly stated as limitations. Alternatively, the authors may want to add a section entitled "Limitations" at the end of the discussion where they list them collectively (as is frequently done in other journals as well.) Since these limitations call for future research, yet another alternative is to combine both, e.g. in a section entitled "Limitations and future studies". In any case the authors should clearly identify these limitations and acknowledge to what extent their conclusions are affected by them and how they might be overcome in the future (which they partially did already).

We have adapted the manuscript to reflect these recommendations. We now highlight this potential alternate explanation for our results in **lines 508-511** and explicitly state it as a shortcoming of the study, making suggestions for how it may be dealt with in future studies in **lines 513-515**.

R3: P 3, L12: *maybe specify the two populations after stating "from two diverse cultural contexts"*

In our research highlights (page 2) we now specify the two cultural contexts by stating "We examine the spontaneous tool use of children living in San Bushmen communities in South Africa and in a large city in Australia.

R3: 7: *maybe explain the "ratchet effect" more clearly and trace it back to Tomasello (1999) (and possibly to Vygotsky (1978) before mentioning subsequent literature (Boyd & Richerson, 2005, Eerkens & Lipo, 2005).*

We have more clearly explained the ratchet effect on **lines 5-9** and now tie it explicitly to Tomasello (1999) and Vygotsky (1978).

Lines 5-9: "The faithful copying of established techniques within a society, particularly by novices and children, combined with the ability to build upon the inventions of others through furthered (copied) innovation and/or copying error (known as the 'ratchet effect'; Tomasello, 1999) are the fundamental drivers of human cumulative culture (Boyd & Richerson, 2005; Eerkens & Lipo, 2005; Vygotsky, 1978)."

R3: 32-34 (and compare 205-207): *it did not become clear to me why invention by 2 participants is considered sufficient to rule out chance. Wouldn't this criterion vary with sample size? Some more justification might be necessary here. The information provided in 267-270 seems to be relevant in this respect as well, but it is not entirely clear to me how.*

We have now more clearly specified why the likelihood of a complex behaviour such as tool use being invented by random chance is unlikely. See **lines 40-41:** "the likelihood that all necessary events required for successful

tool use to emerge by random is very low”. We have also added an additional reference here that supports this criterion (Bandini & Tennie, 2017).

R3: 43: *The cognitive and physical capacities needed for tool use are, among others, addressed in the comprehensive work of Francois Osiurak. A suitable paper in this respect is: Osiurak, F., & Badets, A. (2016). Tool use and affordance: Manipulation-based versus reasoning-based approaches. Psychological Review, 123, 534-568. Although it is not developmental, this paper seems relevant since it includes aspects that are also relevant in development, namely object knowledge (as in manipulation-based approaches) and mechanical knowledge (as in reasoning-based approaches).*

We have added Osiurak & Badets (2016) to the manuscript in **line 52**.

R3: 145-184 (on the 3 Bushmen samples): *I found the characterization of the 3 Bushmen samples quite heterogeneous. It might be better to provide information about the same (selected) criteria, e.g., geographical location, history, SES, setting (of the testing), influence of Westernized society, and, most importantly, aspects relevant for tool-use (playing with traditional toys (manufactured from local materials) or with Western toys). These criteria could also be used for the comparison of the South- African sample (as a whole) with the Australian sample (185-195).*

We believe we have already presented the demographic information we have for each of our communities in the following order – geographic location and setting of testing, history, SES level, influence of Western society, and toy availability. Clear information regarding SES level was not always available in each community, so instead we have provided the qualitative indicators of SES level, such as housing and infrastructure. This content is retained from the first manuscript.

We have added more specification about the testing setting for Africa groups in **lines 178-179**: “recruited within crèches and kindergartens with the !Xun or Khwe communities” and in **lines 194**: “recruited from kindergartens, schools and one community center” to clarify this.

R3: 255: *Were the parents of the Bushmen children also informed after the study, in some way?*

Thank you for highlighting this omission. We have added details to the debriefing procedure for the Bushmen children. As parents were not present at the schools and creches, verbal debriefing was given to their present guardians (the crèche teachers) for dissemination to parents, and debriefing was also verbally delivered to community gatekeepers by the researchers. See **lines 277-280**.

R3: 267: *Besides overall kappa, please provide kappa values for each of the 4 scores (a-d).*

We have added individual kappa ratings for each of the scores, see **lines 294-296**.

R3: 280: *please indicate explicitly where in the supplementary materials this information can be found (Table S6 or S7 (?))*

We now refer to Table S6 and S7 on **lines 309**.

R3: 440: *the information that Australian children could solve the tasks significantly faster on average than Bushmen children (60 s vs 74 s, respectively) should be presented before in the results section or be presented and referred to in the supplementary materials.*

We have moved the result about differing average time of success between the groups to the results section (see **lines 367-369**).

R3: 455: *there is a left-over preposition “in” which should be deleted.*

This preposition has been deleted.

R3: Endnote 4, (P 37, line 29 – p 38, line 7: *As mentioned in my comments to the editor in the section on “ethics” in the online referee system, I would suggest to limit this endnote to the relevant information explaining why it was difficult to obtain consent from the South African San Council (SASC) but leave out (most of) the historical material.*

We have reduced the historical context of the ethical review board notes in **endnote 6 referenced in line 158**, however we retain the information we deem critical to accurately describing the processes we had to undergo to obtain ethical approvals in South Africa.